# Mitochondrial genomes of Pleistocene megafauna retrieved from recent sediment layers of two Siberian lakes

Peter Andreas Seeber[1]*, Laura Batke[1], Yury Dvornikov[2,3], Alexandra Schmidt[1], Yi Wang[1], Kathleen Stoof-Leichsenring[4], Katie Moon[5,6], Samuel H Vohr[7], Beth Shapiro[5,6], Laura S Epp[1]*

[1]Department of Biology, University of Konstanz, Konstanz, Germany; [2]Agroengineering Department/Department of Landscape Design and Sustainable Ecosystems, Agrarian and Technological Institute, RUDN University, Moscow, Russian Federation; [3]Laboratory of Carbon Monitoring in Terrestrial Ecosystems, Institute of Physicochemical and Biological Problems of Soil Science of the Russian Academy of Sciences, Pushchino, Russian Federation; [4]Alfred Wegener Institute Helmholtz Centre for Polar and Marine Research, Polar Terrestrial Environmental Systems, Potsdam, Germany; [5]Department of Ecology and Evolutionary Biology, University of California, Santa Cruz, Santa Cruz, United States; [6]Howard Hughes Medical Institute, University of California, Santa Cruz, Santa Cruz, United States; [7]Embark Veterinary, Inc, Boston, United States

*For correspondence:
seeber.pa@gmail.com
(PAndreasS);
laura.epp@uni-konstanz.de (LSE)

**Abstract** Ancient environmental DNA (aeDNA) from lake sediments has yielded remarkable insights for the reconstruction of past ecosystems, including suggestions of late survival of extinct species. However, translocation and lateral inflow of DNA in sediments can potentially distort the stratigraphic signal of the DNA. Using three different approaches on two short lake sediment cores of the Yamal peninsula, West Siberia, with ages spanning only the past hundreds of years, we detect DNA and identified mitochondrial genomes of multiple mammoth and woolly rhinoceros individuals—both species that have been extinct for thousands of years on the mainland. The occurrence of clearly identifiable aeDNA of extinct Pleistocene megafauna (e.g. >400 K reads in one core) throughout these two short subsurface cores, along with specificities of sedimentology and dating, confirm that processes acting on regional scales, such as extensive permafrost thawing, can influence the aeDNA record and should be accounted for in aeDNA paleoecology.

## eLife assessment

This work presents **convincing** evidence for the presence of wooly mammoth/rhinoceros ancient environmental DNA (aeDNA) far from the time likely to host living individuals: what is effectively a genetic version of a geological inclusion. These are **important** findings that will have ramifications for the interpretation and conclusions extracted from aeDNA more generally.

## Introduction

Sedimentary deposits constitute highly valuable archives of past ecosystem changes as they contain dateable layers with organismic remains including ancient DNA (aDNA). Such remains are typically assumed to represent the ecosystem of the time around which the respective stratum was deposited.

**Table 1.** Sediment cores retrieved from two lakes on the Yamal peninsula, Siberia.

| Lake | Coordinates | m above sea level | Area | Water depth | Core length |
|------|-------------|-------------------|------|-------------|-------------|
| LK-001 | 70°16'45.6" N, 68°53'02.8" E | 28 | 38 ha | 17 m | 80 cm |
| LK-007 | 70°16'02.8" N, 68°59'35.7" E | 36 | 39 ha | 14 m | 75 cm |

aDNA from sediments has yielded remarkable insights regarding paleoecology, phylogeography, and extirpation and extinction events of keystone taxa such as mammoths (*Haile et al., 2009*; *Boessenkool et al., 2012*; *Graham et al., 2016*; *Murchie et al., 2021b*). Based on such ancient environmental DNA (aeDNA), a recent study proposed that the woolly mammoth (*Mammuthus primigenius*) may have survived in Eurasia for much longer than previously assumed, as the authors retrieved mammoth DNA sequences in sediment layers that were approximately 4.6–7 thousand years (kyr) younger than the most recent mammoth fossils *Wang et al., 2021*; however, in response to this interpretation, *Miller and Simpson, 2022* opined that these results may be more likely due to taphonomic processes leading to release of aeDNA from the remains of long-dead organisms from permafrost, where it is well preserved.

Conclusions derived from aeDNA isolated from sediment cores rely on the stratified nature of the remains in question and dating of the respective layer by radiometric methods. However, in theory, various physical and geochemical processes such as translocation of DNA through sediment strata (*Haile et al., 2007*), re-deposition of older sediment carrying DNA of extinct organisms (*Arnold et al., 2011*), and preservation bias (*Boere et al., 2011*) can distort the biological signal of aeDNA and thus bias the accuracy of allocation of taxa to specific time periods (*Armbrecht et al., 2019*). For lake sediments, deposited under aquatic conditions, studies have suggested that leaching is not a concern (*Parducci et al., 2017*), but it has been observed in soils and cave sediments (*Haile et al., 2007*). The question of obtaining last appearance dates of extinct taxa using aeDNA in dated sediment layers (*Haile et al., 2009*) is under discussion, but the surprisingly young records published so far still date to multiple thousands of years before present and thus lie within a timeframe of possible late survival.

## Results and discussion

In 2019, we retrieved short subsurface sediment cores from two Arctic thermokarst lakes (LK-001 and LK-007, located approximately 5 km apart, over massive permafrost; *Table 1*; *Dvornikov et al., 2016*) on the Yamal peninsula, Siberia, to extract DNA and assess changes in mammal abundances in the Arctic over the past decades and centuries. From lake LK-001, we collected a secondary core which was sliced in the field at 1 cm steps for Pb$^{210}$ radiometric dating, which indicated that the sediments at the top of this core were deposited recently, and that the core spanned the past few centuries (*Appendix 1—table 7*). The cores for DNA extraction were closed in the field immediately after retrieval and were then transported to the dedicated aDNA laboratories of the University of Konstanz, Germany. In this lab facility, established in 2020, no other samples from the Arctic or from any large mammals had been processed previously. From core LK-001, we isolated DNA and produced genomic double-stranded libraries from 23 samples, from 1.5 to 80 cm core depth (Supplement section 1), according to standard procedures. The core was opened and all subsequent steps until index PCR setup were carried out under customary aDNA laboratory conditions. In particular, the core opening facilities and the lab are located in buildings separated from the downstream molecular genetic analyses, the ventilation of the aDNA lab is based on a HEPA filter system and positive air pressure, and the lab is subjected to nightly UV radiation. Work in the lab is conducted under strict aDNA precautions, adhering to established aDNA protocols (*Fulton and Shapiro, 2019*). We enriched the libraries for mammalian DNA using a custom RNA bait panel produced from complete mitogenome sequences of 17 mammal species that currently or previously occurred in the Arctic (adapted from *Murchie et al., 2021a*). The enriched libraries were sequenced, and we mapped the sequences against a database of 73 mammal mitogenomes, followed by BLASTn alignment against the complete NCBI nt database. We thus retrieved mitogenomic sequences of mammals that were expected during the age range covered by the core (*Appendix 1—table 9*), for example, reindeer (*Rangifer tarandus*), Arctic lemming (*Dicrostonyx torquatus*), and hare (*Lepus*); however, throughout the entire core, there were abundant sequences of two species that have been extinct for several thousand years, that

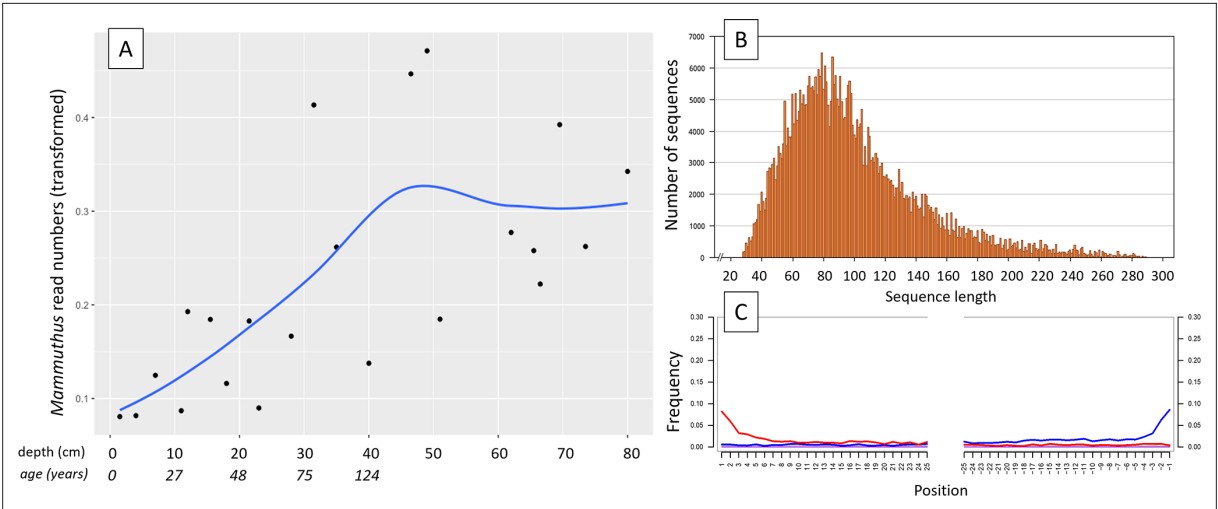

**Figure 1.** aDNA of *Mammuthus* in recent lake sediments. (**A**) Read counts assigned to *Mammuthus* (square-root-transformed proportion of the respective number of raw reads per library) after hybridization capture enrichment of aeDNA of core LK-001 (shown are results of 22 libraries; one library was excluded as it did not produce any reads assigned to mammals); square-root transformation of percentage. Indicated are sample depths (in cm; 1.5–80 cm) and approximate ages as per [210]Pb chronology (*Appendix 1—table 7*; to a maximum depth of 39.5 cm). The solid line indicates the general trend. Across the 22 libraries: (**B**) Fragment length distribution and (**C**) damage patterns (red indicates C-to-T transitions, blue G-to-A transitions. the Y-axis indicates the percentage of positions with a nucleotide change, the X-axis indicates the position along the fragment).

is mammoth (*Mammuthus primigenius*) and woolly rhinoceros (*Coelodonta antiquitatis*). Twenty-two of the 23 LK-001 libraries produced >1000 reads, each, assigned to *Mammuthus*, with read counts ranging from 2852 to 72,919 (mean 21,140±17,296). Negative controls (extraction and library blanks) did not produce any reads assigned to mammals. In the sample with the highest *M. primigenius* read counts (31.5 cm depth, dated to 81 years), the coverage of the reference mitogenome (NCBI accession NC_007596.2) was 95.3%, (434 (±213)-fold). Across all samples, 465,080 reads assigned to mammoth were produced, with 98.3% coverage (2762 to ±1176 fold). Read lengths ranged from 28 to 289 bp (mean 100±44 bp; *Figure 1*). The number of woolly mammoth reads decreased from lower samples towards the top of the core (*Figure 1*). Signatures of post-mortem DNA decay were comparably minor (*Figure 1*), with reference to an *M. primigenius* genome downloaded from NCBI (accession NC_007596.2), and mapping suggested that the sequences throughout the core originated from multiple individuals. Further analyses of the three libraries with the most mammoth reads

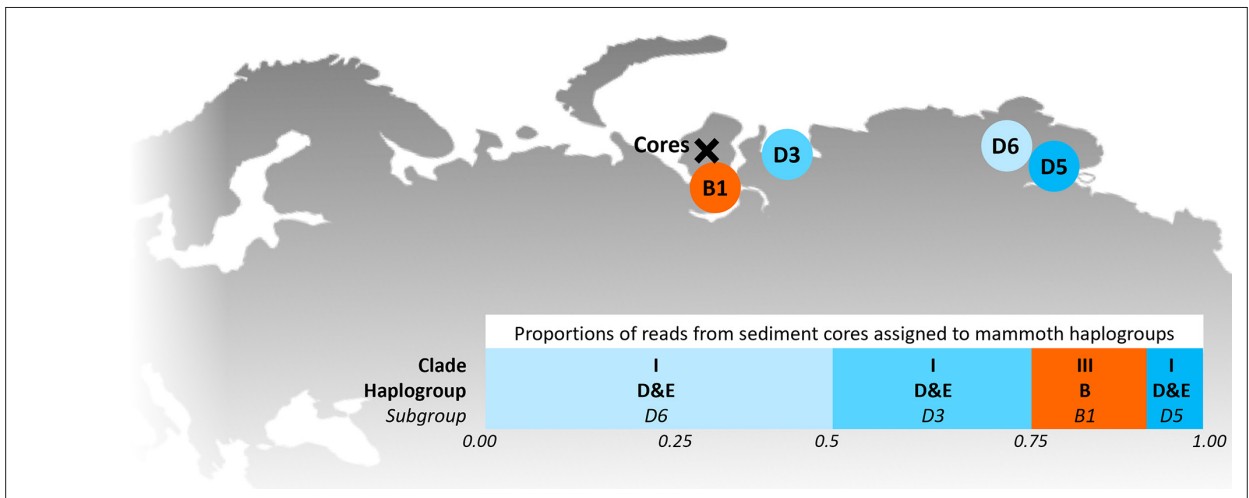

**Figure 2.** Locations of the sediment cores of the present study (Yamal peninsula, Siberia) and previously retrieved mammoth remains and their haplo(sub)groups (*Appendix 1—table 6*). The bar chart indicates a maximum-likelihood estimate of the haplogroup proportions derived from the reads from the three sediment core libraries with the most mammoth reads.

using mixemt (*Vohr et al., 2017*) identified a number of mitochondrial haplogroups in the sequences from the core, suggesting that they originated from a multiple individuals (*Figure 2*). The haplogroups identified were known to occupy the region, and it seems likely the sequences reflect a history of mammoth occupation at the core site. Twelve of the 23 libraries produced >100 reads, each, assigned to woolly rhinoceros, with a total of 2737 reads and a cumulative coverage of 44% (assembled to NC_012681.1 2). Further analyses are mostly focused on the mammoth sequences as these occurred in substantially higher numbers.

As these results were entirely incongruent with the temporal occurrence of these two species, we employed several methods to confirm their validity, that is conventional PCR and Sanger sequencing of a mammoth cytochrome c oxidase subunit I (COI) fragment, mammal metabarcoding, and droplet digital PCR (ddPCR) of a mammoth cytochrome b fragment. We performed these analyses on core LK-001 and, to evaluate whether this is a locally isolated phenomenon, on a second short core, LK-007, from a nearby lake, which had been opened and processed under the same conditions indicated above to prevent contamination. Additionally, we screened the core LK-001 for plant macrofossils, of which three were sent for [14]C AMS dating.

Conventional PCR and Sanger sequencing confirmed amplification of a COI fragment of *M. primigenius*. Mammal metabarcoding produced mammoth sequences (74 bp) in 13 samples of core LK-001 and 9 samples of core LK-007. In the LK-001 core, the highest *M. primigenius* read count occurred at 31.5 cm (2,992 reads), and in the LK-007 core at 26 cm *M. primigenius* (3,580 reads). ddPCR produced *M. primigenius* sequences in 14 samples of each core. Metabarcoding and ddPCR patterns across the cores were similar, although not completely congruent, as ddPCR appeared to be more sensitive (Appendix 1—figure 1). The dates retrieved by radiocarbon dating were not congruent with the initial age inference suggested by Pb[210]. While the lowest and topmost sample (with ages of 1547±228 and 339±79 uncal. yrs BP respectively) suggest relatively young ages and agree in their temporal succession, the middle sample, at 51 cm, yielded a radiocarbon age of 8677±132 years.

The mammoth was abundant throughout most of Eurasia during the Pleistocene, but populations declined at the end of the Pleistocene, with the species going extinct in the mid-Holocene (*Nogués-Bravo et al., 2008*). The youngest fossils from mainland Siberia have been dated to 9650 years (*Stuart et al., 2002*). The woolly rhinoceros was a cold-adapted megaherbivore, which was abundant from western Europe to north-east Siberia during the Middle to Late Pleistocene (*Kahlke and Lacombat, 2008*). This species was predominantly grazing, probably resorting to browsing only due to seasonal vegetation restrictions (*Kahlke and Lacombat, 2008*; *Rey-Iglesia et al., 2021*; *Stefaniak et al., 2021*). The reasons underlying the extinction of this species at ca 14 ka BP are not entirely clear, but it is largely attributed to sudden climate warming and subsequent habitat unsuitability due to vegetation changes (*Stuart and Lister, 2012*; *Lord et al., 2020*; *Wang et al., 2021*), likely coupled with human influence (*Fordham et al., 2022*), in the Bølling-Allerød interstadial warming.

The present data, which implies frequent and abundant Pleistocene megafaunal DNA throughout a sediment core deposited in the lake over the past centuries suggests that physical processes, rather than presence of live organisms, are responsible for the recovery of this DNA. While not in itself fully conclusive, our data suggests the source of the DNA of Pleistocene mammals from older permafrost deposits, either from a carcass or from sedimentary materials carrying the DNA. The numbers of mammoth sequences increased with depth downcore, with comparably low abundances over the most recent 11 cm of the core (aged <30 years), pointing to a decrease of input of this DNA through time. The apparently rather limited extent of aDNA damage in the mammoth sequences suggests that the source of this DNA has been preserved exceptionally well, which also suggests an origin from permafrost, and the specific dynamics of thawing and re-deposition of material in the study area offer an explanation.

Here, the active thermokarst likely began during the climatic optimum of the Holocene (9000–3,000 BC; *Savvichev et al., 2021*). The LK-001 and LK-007 lake basins are interbedded mainly into the IV[th] marine plain, formed in the Kazantsevskaya, Marine Isotopic Stage 5. The permafrost of these lake catchments was formed no earlier than 70—60 kyr BP after the Kazantsevskaya transgression in more sub-aerial conditions and with mean annual temperatures 6 to 7 °C lower than modern temperatures (*Baulin et al., 1981*). As this area was under coastal-marine conditions for a long period, these lake basins may be paleo-marine remnants, or they were formed later as a result of thermokarst over the segregated or tabular (massive) ground ice during the Holocene climatic optimum (*Kachurin, 1961*).

The area is also subject to abrupt permafrost thaw (thermo-denudation), resulting, for example, in the formation of retrogressive-thaw slumps (thermocirques) and the transport of a large amounts of thawed terrestrial material into the lake water (*Dvornikov et al., 2018*). Such abrupt permafrost thaw processes normally appear adjacent to lakes and can form specific geomorphological elements, that is, thermo-terraces (*Kizyakov et al., 2006*) within lake catchments and lakes; they are normally polycyclic processes appearing due to the extension of a seasonally thawed layer (active layer) up to the top of massive ice and more favorable thermal conditions within the existing forms (*Leibman and Kizyakov, 2007*; *Kokelj et al., 2009*). Traces of past thermo-denudation can be observed within both lake catchments. In catchments of five neighboring lakes, large retrogressive-thaw slumps appeared in recent years (2012–2013) accompanied by the thaw and lateral transport of modern and Late Pleistocene deposits into lakes. Additionally or alternatively, thermo-erosion of upper geomorphological levels and transport via stream networks could transport ancient material into the modern lacustrine sediments. However, the two studied lakes are headwater lakes (with outlet, no apparent inlet) and this option can only be considered in terms of small thermo-erosional valleys within the catchments.

An alternative mechanism for the redistribution of Late Pleistocene material in the sediments is related to subcap methane emission (bubbling) from degrading permafrost beneath the lake bottom. In-lake bubbling can be observed in a circum-Arctic scale: in North-East Siberia, Alaska and Canada. This is common especially in lakes with a depth exceeding two meters, which do not freeze entirely up to the bottom in winter, leading to the formation of a talik (a layer of year-round unfrozen ground that lies in permafrost area). The expansion of the talik may further trigger subcap methane emission, which can reach 40–70 kg yr$^{-1}$ of pure (94–100%) methane in neighboring lakes (*Kazantsev et al., 2020*). The constant methane seepage does not allow ice to be formed in winter (whereas the normal winter ice thickness is approximately 1.5 m) and can potentially disturb the stratigraphy of lake sediments. Additionally, dramatic emissions of methane can form craters in terrestrial and lacustrine environments (*Dvornikov et al., 2019*). In this case, Late Pleistocene sediments will well be re-distributed within the water-body and the entire stratigraphy will be mixed.

In the case reported here, the simultaneous finding of a Pb$^{210}$ chronology indicating recent sediment deposition and of plant macrofossils that dated to >8000 years BP in a sample from 36.5 cm, suggest lateral input of ancient material, including the mammalian DNA, putatively related to permafrost thawing processes. Numerous studies on aDNA discussed possible leaching through sedimentary strata of the DNA itself, yet it was typically considered not an issue as most of these studies were conducted under stable permafrost or similar conditions (e.g. *Hansen et al., 2006*; *Haile et al., 2007*, but see *Andersen et al., 2012*). Permafrost thawing and re-deposition of material adds a new dimension to this problem of temporal interpretation. The fact that we retrieved mammoth sequences from both cores of two lakes located approximately 5 km apart suggests that this is not an isolated phenomenon but occurs on a regional or even larger scale. Given the wide spread of abrupt permafrost thaw processes in the Arctic plains (West Siberia, Taimyr, Chukotka, Alaska, Canadian Arctic; *Kizyakov et al., 2006* and references therein), the phenomena of disturbed stratigraphy of lacustrine sediments can potentially be observed at a pan-Arctic scale. While this indicates that temporal interpretation of sedimentary aeDNA records should be exercised with caution, our study also demonstrates that a careful evaluation of available information on the site and ecosystem in conjunction with the use of independent dating techniques can uncover incongruencies. This is more difficult in older time periods, where artefactual stratigraphies caused by equivalent processes acting for a limited time will not be detected as easily as in our case of long extinct species. The same applies to extant taxa or those which have undergone extinction or extirpation more recently, the presence of which cannot as easily be excluded as in the current example. However, we suggest that the inclusion of robust dating techniques and knowledge of local geophysical processes can provide good arguments to evaluate the reliability of aeDNA records.

## Materials and methods
### Field sites, DNA isolation and hybridization capture enrichment
In 2019, sediment cores (6 cm diameter; *Table 1*) were retrieved from two lakes (LK-001 and LK-007, respectively, *Table 1*) on the Yamal peninsula, Siberia, using a UWITEC piston corer (UWITEC, Mondsee, Austria). The lakes were located approximately 5 km apart. The cores were transported to

the aDNA laboratories of the University of Konstanz, Germany; from lake LK-001, a secondary core was taken which was sliced in the field at 1 cm steps for radiometric dating from 0 to 39.5 cm depth, performed at the Environmental Radioactivity Research Centre, University of Liverpool (Supplement section 9). Additional 14 C dating of three specimens of plant remains, extracted at 36.5 cm, 51 cm, and 74 cm, was performed (Supplement section 9).

Sedimentary DNA was isolated from 23 samples of core LK-001 and from 16 samples of core LK-007 using commercially available kits with modified protocols (Supplement section 1). The extracts of core LK-001 were subjected to library preparation for capture enrichment. Enrichment probes were designed from mitogenomes of 17 herbivorous mammal species that currently or previously occurred in the Arctic (*Appendix 1—table 2*) and few lichen sequences (*Appendix 1—table 3*). Genomic libraries were produced according to *Li et al., 2013* with some modifications (*Seeber et al., 2019*; *Seeber et al., 2023*; *Li et al., 2023*; Supplement section 2). Filtered reads were mapped to mammalian mitogenomes, followed by BLASTn alignment against the complete NCBI nucleotide database and subsequent metagenomic analyses using MEGAN (*Huson et al., 2016*). Reads assigned to *Mammuthus* were mapped to a complete *M. primigenius* reference mitogenome (NCBI accession NC_007596.2); reads assigned to *Coelodonta antiquitatis* were mapped to the NCBI reference mitogenome NC_012681.1 2. The reads mapped to mammoth from the top three libraries were assigned to haplogroup using mixemt (https://github.com/svohr/mixemt, Copy archieved at *Vohr, 2024*) with a custom-made representative panel of 15 mammoth mitogenomes (*Figure 2*; *Appendix 1—table 6*).

## Conventional PCR, mammal metabarcoding, and ddPCR

Based on the enriched fragments with the highest coverage, PCR primers specific to *M. primigenius* were designed using Geneious Prime 2022.1.1 (*Kearse et al., 2012*), that is mamm801 (5`- CCCA TGCAGGAGCTTCTGTAGA-3`) and mamm800r (5`-GACATAGCTGGAGGTTTTATGT-3`) to produce a 121 bp amplicon of the CO1 gene. The specific PCR conditions are described in Supplement section 6. Mammal metabarcoding PCRs were performed on 21 DNA extracts of core LK-001 and 27 samples of core LK-007, with eight independent replicates, each. Each batch of PCRs included one non-template control. Established metabarcoding primers were used (*Giguet-Covex et al., 2014*), and human blocking primers (*Garcés-Pastor et al., 2021*) were included. PCR conditions are described in the supplementary material. Sequencing was performed on an Illumina NovaSeq platform, with 2x150 reads. The raw data were processed as described in the supplement. We used the sample LK-001_66.5 of core LK-001 which had produced one of the highest *Mammuthus* read counts after enrichment. The specific target amplified by the primer pair were used to design a probe (5`-GGAT ACTCCTGCAAGGTGAAGTG-3`). With this probe, ddPCR was performed with 24 extracts of core LK-001 and with 30 extracts of core LK-007, with three replicates, each (Supplement section 8).

## Acknowledgements

This research was funded through the 2017–2018 Belmont Forum and BiodivERsA joint call for research proposals, under the BiodivScen ERA-Net COFUND program, and with the funding organizations Deutsche Forschungsgemeinschaft (DFG grant EP-98/3–1 to LSE), Agence Nationale de la Recherche (ANR), Research Council of Norway (NFR), the Swedish Research Council for Environment, Agricultural Sciences and Spatial Planning (Formas), Academy of Finland, National Science Foundation (NSF) and the Natural Sciences and Engineering Research Council of Canada (NSERC-CRSNG). Field work was enabled through a grant of the Young Scholar Fund of the University of Konstanz to LSE. YD was supported by the RUDN University Strategic Academic Leadership Program. Expedition logistics were supported by the Department of Science and Innovations of Yamalo-Nenets Autonomous Okrug and the Non-profit organization «Russian Center of Arctic Exploration». We thank PD Dr. Elena Marinova for assistance and discussions concerning dating of plant macrofossils and Patrick Bartolin for assistance in the wet lab.

## Additional information

### Competing interests

Peter Andreas Seeber: Employed by and owns stock of Thermo Fisher Scientific. Samuel H Vohr: Employed by Embark Veterinary, Inc. The other authors declare that no competing interests exist.

## Funding

| Funder | Grant reference number | Author |
| --- | --- | --- |
| Deutsche Forschungsgemeinschaft | EP-98/3-1 | Laura S Epp |
| Biodiversa+ | BiodivScen ERA-Net COFUND | Laura S Epp |

The funders had no role in study design, data collection and interpretation, or the decision to submit the work for publication.

## Author contributions

Peter Andreas Seeber, Conceptualization, Data curation, Formal analysis, Validation, Investigation, Visualization, Methodology, Writing – original draft, Writing – review and editing; Laura Batke, Formal analysis, Methodology; Yury Dvornikov, Beth Shapiro, Writing – review and editing; Alexandra Schmidt, Yi Wang, Formal analysis; Kathleen Stoof-Leichsenring, Resources; Katie Moon, Samuel H Vohr, Formal analysis, Writing – review and editing; Laura S Epp, Supervision, Funding acquisition, Writing – original draft, Project administration, Writing – review and editing

## Author ORCIDs

Peter Andreas Seeber ⓘ http://orcid.org/0000-0003-4401-8248
Alexandra Schmidt ⓘ http://orcid.org/0000-0001-9262-0941
Kathleen Stoof-Leichsenring ⓘ http://orcid.org/0000-0002-6609-3217
Laura S Epp ⓘ https://orcid.org/0000-0002-2230-9477

Reviewer #2 (Public Review): https://doi.org/10.7554/eLife.89992.3.sa1
Reviewer #3 (Public Review): https://doi.org/10.7554/eLife.89992.3.sa2
Author Response https://doi.org/10.7554/eLife.89992.3.sa3

---

# Additional files

## Supplementary files

• MDAR checklist

## Data availability

Sequence data of the hybridization capture were made available as an NCBI BioProject (PRJNA1082062).

The following dataset was generated:

| Author(s) | Year | Dataset title | Dataset URL | Database and Identifier |
| --- | --- | --- | --- | --- |
| Seeber PA | 2024 | Mitochondrial genomes of Pleistocene megafauna retrieved from recent sediment layers of two Siberian lakes | https://www.ncbi.nlm.nih.gov/bioproject/PRJNA1082062 | NCBI BioProject, PRJNA1082062 |

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

## Appendix 1

### Sampling and DNA isolation

The sediment cores were subsampled according to procedures described by *Epp et al., 2019*, extracting approximately 1–2 g sediment from the center of the core at close depth intervals (*Appendix 1—table 1*). Sampling was performed in a window-less cold room at 4 °C, adhering to standard aDNA precautions. DNA was extracted using two highly similar kits. Sample depths and respective extraction protocols for both cores are listed below (*Appendix 1—table 1*).

**Appendix 1—table 1.** Sample depths and extraction protocols.

| (a) LK-001 | | (b) LK-007 | |
|---|---|---|---|
| Sample depth (cm) | Extraction protocol* | Sample depth (cm) | Extraction protocol* |
| 1.5 | | 2.0 | PowerLyzer |
| 4.0 | | 4.0 | PowerSoil Pro |
| 7.0 | | 6.0 | PowerLyzer |
| 11.0 | | 8.0 | PowerSoil Pro |
| 12.0 | | 10.0 | PowerLyzer |
| 15,5 | | 12.0 | PowerSoil Pro |
| 18.0 | | 14.0 | PowerSoil Pro |
| 21.5 | | 18.0 | PowerLyzer |
| 23.0 | | 20. | PowerSoil Pro |
| 28.0 | | 24.0 | PowerLyzer |
| 31.5 | | 26.0 | PowerSoil Pro |
| 35.0 | PowerLyzer | 30.0 | PowerSoil Pro |
| 40.0 | | 32.0 | PowerSoil Pro |
| 44.0 | | 36.0 | PowerLyzer |
| 46.5 | | 38.0 | PowerSoil Pro |
| 49.0 | | 42.0 | PowerSoil Pro |
| 51.0 | | 44.0 | PowerLyzer |
| 62.0 | | 48.0 | PowerLyzer |
| 65.5 | | 50.0 | PowerLyzer |
| 66.5 | | 54.0 | PowerLyzer |
| 69.5 | | 56.0 | PowerLyzer |
| 73.5 | | 60.0 | PowerLyzer |
| 80.0 | | 62.0 | PowerSoil Pro |
| | | 64.0 | PowerLyzer |
| | | 66.0 | PowerSoil Pro |
| | | 68.0 | PowerLyzer |
| | | 70.0 | PowerSoil Pro |

*Kits from Qiagen (Hilden, Germany).

The PowerLyzer DNA extraction was performed according to a previous study (*Alsos et al., 2020*), with some modifications (*Appendix 1—table 2*).

**Appendix 1—table 2.** PowerLyzer DNA extraction.

| | |
|---|---|
| *Day 1:* | 1. Add 750 µL PowerBead Solution to the PowerBead Tube<br>2. Transfer 0.25–0.35 g manually homogenized sediment to PowerBead Tube<br>3. FastPrep bead beating: two times Quickprep protocol (20 s at 4.0 m/s); briefly centrifuge to eliminate foam<br>4. Lysis mix (per sample):<br>Solution C1 60 µL<br>Proteinase K (20 mg/mL) 2 µL<br>1 M DTT 25 µL<br>Vortex and briefly centrifuge.<br>5. Add 87 µL lysis mix to each PowerBead Tube; vortex for 5 min; invert the tube, flick and vortex to dissolve pellet, if present.<br>6. Incubate overnight at 56 °C and 12 rpm |
| *Day 2:* | 1. Remove PowerBead Tube from the incubator oven and allow to cool to room temperature<br>2. Centrifuge at 10,000 x $g$ for 1 min<br>3. Add 250 µL Solution C2 to a 2 mL collection tube<br>4. Avoiding the pellet, pour the supernatant from step 2 into the collection tube containing Solution C2; vortex for 5 s<br>5. Incubate at 2–8 °C for 10 min<br>6. Centrifuge at 10,000 x $g$ for 1 min<br>7. Label a new clean 2 mL collection tube and add 250 µL Solution C3<br>8. Avoiding the pellet, pour up to 800 µL of supernatant to the collection tube containing Solution C3; vortex for 5 s<br>9. Incubate at 2–8 °C for 10 min<br>10. Centrifuge at 10,000 x $g$ for 1 min<br>11. Label a new clean 5 mL collection tube and add 1,400 µL Solution C4<br>12. Avoiding the pellet, pour 880 µL of supernatant to the collection tube containing Solution C4; vortex for 5 s; briefly centrifuge<br>13. Load 650 µL of the Solution C4-supernatant mix on a Spin Column<br>14. Centrifuge the Spin Column at 10,000 x $g$ for 1 min<br>15. Transfer the Spin Column to a new 2 mL collection tube<br>16. Repeat steps above until all Solution C4-supernatant mix has been loaded onto the Spin Column<br>17. Centrifuge the Spin Column at 10,000 x $g$ for 1 min<br>18. Load 500 µL of Solution C5 on the Spin Column<br>19. Centrifuge the Spin Column at 10,000 x $g$ for 1 min<br>20. Transfer the Spin Column to a new 1.5 mL collection tube<br>21. Centrifuge the Spin Column at 10,000 x $g$ for 1 min<br>22. Transfer the Spin Column to a new, labelled, and sterile 1.5 mL collection tube<br>23. Add 65 µL of elution buffer to the center of the filter membrane<br>24. Incubate at room temperature for 10 min<br>25. Centrifuge the Spin Column at 10,000 x $g$ for 1 min<br>26. Repeat steps above for a final elution volume of 120–125 µL |

PowerSoil Pro DNA extraction was performed as follows:

| | |
|---|---|
| *Day 1:* | 1. Spin the PowerBead Pro Tube briefly; add up to 500 mg soil and 800 µL Solution CD1. Vortex briefly to mix. Add 20 µL Proteinase K (2 mg/mL).<br>2. FastPrep bead beating: two times 20 s at 4.0 m/s; briefly centrifuge to eliminate foam. Incubate at 56 °C overnight. |
| *Day 2:* | 3. Centrifuge the PowerBead Pro Tube at 15,000 x $g$ for 1 min.<br>4. Transfer the supernatant to a clean 2 mL microcentrifuge tube.<br>5. Add 200 µL Solution CD2 and vortex for 5 s.<br>6. Centrifuge at 15,000 x $g$ for 1 min at room temperature. Transfer up to 700 µL supernatant to a clean 2 mL microcentrifuge tube.<br>7. Add 600 µL Solution CD3 and vortex for 5 s.<br>8. Load 650 µL of the lysate on an MB Spin Column and centrifuge at 15,000 x $g$ for 1 min.<br>9. Discard the flow-through and repeat step 8 to ensure that all of the lysate has passed through the MB Spin Column.<br>10. Place the MB Spin Column in a clean 2 mL collection tube.<br>11. Add 500 µL Solution EA to the MB Spin Column. Centrifuge at 15,000 x $g$ for 1 min.<br>12. Discard the flow-through and place the MB Spin Column back into the same collection tube.<br>13. Add 500 µL Solution C5 to the MB Spin Column. Centrifuge at 15,000 x $g$ for 1 min.<br>14. Discard the flow-through and place the MB Spin Column in a new 2 mL collection tube<br>15. Centrifuge at 16,000 x $g$ for 2 min. Place the MB Spin Column in a new 1.5 mL elution tube.<br>16. Add 50–100 µL Solution C6 to the center of the white filter membrane; incubate for 5 min, and centrifuge at 15,000 x $g$ for 1 min. |

## Preparation of genomic libraries for hybridization capture

Double-stranded libraries were prepared from genomic DNA according to the protocol below and were visualized by gel electrophoresis. Each batch of libraries (N=12) was processed with a library blank to control for contamination during library prep. Indexing was performed using individual combinations of P5/P7 index primers.

**Appendix 1—table 3.** Library preparation.

| End repair | per library |
|---|---|
| Mix following components in a sterile low-binding PCR tube | |
| NEBNext End Repair Buffer (10 X) | 5 µL |
| NEBNext End Repair Enzyme Mix | 2.5 µL |
| genomic DNA | 42.5 µL |
| Incubate in a thermal cycler for 30 mins at 20 °C. Purify using QIAquick/MinElute PCR purification kit. Elution: add 32 µl buffer EB and incubate at 37 °C for 5 min before spinning down the DNA at 13,000 rpm for 1 min. | |
| **Adapter ligation** | |
| Mix following components in a sterile low-binding PCR tube | |
| Quick Ligation Reaction Buffer (10 X) | 10 µL |
| nuclease-free water | 4 µL |
| P5/P7 adapter mix (50 µM stock) | 1 µL |
| DNA as purified in step above | 30 µL |
| Quick T4 DNA ligase | 4.8 µL |
| *final adapter concentrations for ancient samples should be between 0.25–0.5 µM. **it is vital to add ligase <u>after</u> mixing DNA with adaptors. | |
| Incubate for 15 min at 25 °C; purify using a QIAquick/MinElute PCR purification kit. Elution: 42 µL buffer EB and incubate at 37 °C for 5 min before spinning down the DNA at 13,000 rpm for 1 min. | |
| **Fill-In Reaction** | |
| Add the following reagents into a low-binding PCR tube | |
| ThermoPol Reaction Buffer | 5 µL |
| dNTPs (10 mM) | 2 µL |
| Bst DNA polymerase | 3 µL |
| DNA as eluted above | 40 µL |
| Incubate: 20 mins at 65 °C 20 mins at 80 °C No purification is needed after this step. | |

**Appendix 1—table 4.** Indexing PCR.

| Reagent | µL |
|---|---|
| H$_2$O | 3.45 |
| Platinum Hifi Taq Buffer 10 X (Thermo Fisher Scientific) | 2.50 |
| dNTPs (25 mM) | 0.25 |
| bovine serum albumin (New England Biolabs) | 1.00 |
| MgSO$_4$ (50 mM) | 1.00 |
| Platinum HiFi (5 U/µL; Thermo Fisher Scientific) | 0.20 |

*Appendix 1—table 4 Continued on next page*

*Appendix 1—table 4 Continued*

| Reagent | µL |
|---|---|
| total | 8.40 |
| Index P5 (10 µM) | 0.80 |
| Index P7 (10 µM) | 0.80 |
| template DNA | 15.00 |

| Thermocycling | | |
|---|---|---|
| °C | t | |
| 94 | 1 min | |
| 94 | 15 s | 8 cycles |
| 60 | 20 s | |
| 68 | 60 s | |
| 68 | 3 min | |
| 20 | store | |

Thereafter, libraries were pooled (four libraries/pool) and were purified using a MinElute PCR purification kit (Qiagen; elution: 2x20 µL). Concentrations were measured using a Qubit device (broad-range assay; Thermo Fisher Scientific).

## Hybridization capture enrichment and sequencing

To design RNA oligonucleotide baits for hybridization capture, we compiled the mitochondrial genome sequences of 17 selected mammal species (comprising 282,168 nucleotides [nt]; *Appendix 1—table 2*) and ITS-1 and ITS-2 sequences of *Cladonia rangiferina* (8546 and 4750 nt, respectively; *Appendix 1—table 3*). The compiled mitochondrial sequences were submitted to Daicel Arbor Biosciences for custom design of 70 bp baits at threefold tiling, collapsed at 99% identity and 85% overlap, resulting in 9339 unique baits (747,120 nt).

Each purified library pool containing four libraries was subjected to one hybridization capture reaction, apart from libraries produced from extraction blanks (N=15), library blanks (N=15), and PCR non-template controls (N=30), all of which were pooled in one reaction due to the minute DNA concentrations (mostly below detection range).

Hybridization capture was performed according to the MYbaits protocol v. 5.00 (Daicel Arbor Biosciences) with the following modifications: a total amount of 150 ng baits per capture reaction was used, and library pools were incubated for hybridization with the baits at 58 °C for 24 hr. Post-capture, the enriched libraries were removed from the beads and were amplified similarly to the PCR protocol used for the indexing PCR and with primers IS5/IS6 (Illumina, San Diego, CA, USA) and 14 cycles. PCR products were then purified using the MinElute PCR Purification Kit (Qiagen), and the final library concentration was measured on an Agilent Bioanalyzer. For sequencing, the enriched libraries were pooled at equal concentrations, and two library pools (produced from 57 and 110 libraries, respectively) were sequenced on an Illumina NovaSeq platform using three Novaseq SP 2x150 flow cells.

**Appendix 1—table 5.** Mitogenome templates of 17 mammal species for design of RNA baits.

| Order | Species | NCBI accession | # of baits |
|---|---|---|---|
| | *Bison bison* | NC_012346.1 | 446 |
| | *Bos primigenius* | NC_020746.1 | 456 |
| | *Saiga tatarica* | NC_013996.1 | 538 |
| | *Ovis canadensis* | NC_015889.1 | 522 |
| Artiodactyla | *Ovibos moschatus* | NC_020631.1 | 542 |
| | *Cervus elaphus* | NC_007704.2 | 523 |
| | *Rangifer tarandus* | NC_007703.1 | 526 |
| | *Alces alces* | NC_020677.1 | 520 |
| | *Camelus ferus* | NC_009629.2 | 583 |
| | *Equus przewalskii* | NC_024030.1 | 575 |
| Perissodactyla | *Coelodonta antiquitatis* | NC_012681.1 | 571 |
| | *Lepus arcticus* | NC_044769.1 | 586 |
| Lagomorpha | *Ochotona collaris* | NC_003033.1 | 591 |
| Proboscidea | *Mammuthus primigenius* | NC_007596.2 | 588 |
| Eulipotyphla | *Sorex tundrensis* | NC_025327.1 | 584 |
| | *Castor canadensis* | NC_033912.1 | 584 |
| Rodentia | *Dicrostonyx torquatus* | NC_034646.1 | 575 |
| 9,310 | | | |

**Appendix 1—table 6.** *Cladonia rangiferina* sequences for RNA bait design (shown are the NCBI accessions).

| ITS-1 |
|---|
| MN756840.1; DQ394367.1; JQ695919.1; MK179592.1; KP031549.1; KP001202.1; AF458306.1; KT792792.1; MK811970.1; KT792788.1; MK508944.1; GU169225.1; KP001197.1; KP001201.1; MK812260.1; MK811708.1; KY119381.1; MK508952.1; KT792789.1; EU266113.1; KY266884.1; KP001192.1; KP001190.1; JQ695918.1; KT792790.1; JQ695920.1; KP001191.1; AF458307.1; KP001200.1; MK508943.1; MK812460.1; MK508937.1; KT792791.1 |
| *resulting in 23 baits* |

| ITS-2 |
|---|
| KT792789.1; KP001190.1; AF458307.1; JQ695919.1; MK179592.1; DQ394367.1; KP001194.1; KP001193.1; KY266884.1; KP001199.1; KP001192.1; MK300750.1; MN756487.1; MK508937.1; KP001200.1; MK812460.1; MK508943.1; KP001191.1; KP031549.1; KP001202.1; AF458306.1; MK811970.1; KY119381.1; KP001198.1; MK812260.1; MK811708.1; GU169225.1; JQ695918.1; KP001201.1 |
| *resulting in 6 baits* |

## Bioinformatics of capture enrichment data

Adapter sequences were removed and sequences were filtered using leeHom with the default ancient DNA settings (*Renaud et al., 2014*). Duplicates were removed, and filtered reads were mapped against a database of 73 mammal mitogenomes (*Appendix 1—table 4*) using BWA v. 0.7.17 (*Li and Durbin, 2009*) and were blastn-aligned against the complete NCBI Genbank nucleotide database (*Benson et al., 2009*; retrieved March 14 2022) with a maximum e-value of 0.01. All blast output files were processed using MEGAN Community Edition 6.19.8 using a weighted LCA algorithm (*Huson et al., 2016*). aDNA degradation was examined using mapDamage 2.2.1 (*Ginolhac et al., 2011*) against a reference genome of *M. primigenius* (NC_007596.2). The respective command lines are shown in *Appendix 1—table 5*. The reads blast-assigned to *M. primigenius* were mapped to the complete *M. primigenius* reference mitogenome to produce consensus sequences using Geneious Prime 2023.0.1 (*Kearse et al., 2012*).

**Appendix 1—table 7.** Reference mitogenomes used for mapping.

| | | | |
|---|---|---|---|
| NC_020679.1 | *Antilocapra americana* | NC_018783.1 | *Equus ovodovi* |
| NC_012346.1 | *Bison bison* | HM118851.1 | *Equus hemionus* |
| NC_020746.1 | *Saiga tatarica* | MK982180.1 | *Equus asinus* |
| NC_013996.1 | *Bos primigenius* | NC_012681.1 | *Coelodonta antiquitatis* |
| NC_015889.1 | *Ovis canadensis* | NC_007596.2 | *Mammuthus primigenius* |
| NC_020630.1 | *Oreamnos americanus* | FR691686.1 | *Castor fiber* |
| NC_020631.1 | *Ovibos moschatus* | NC_033912.1 | *Castor canadensis* |
| NC_027233.1 | *Bison priscus* | NC_034313.1 | *Dicrostonyx groenlandicus* |
| NC_009629.2 | *Camelus ferus* | NC_034646.1 | *Dicrostonyx torquatus* |
| KR822422.1 | *Camelops cf. hesternus* | JN181159.1 | *Peromyscus leucopus* |
| NC_013836.1 | *Cervus elaphus xanthopygus* | NC_006853.1 | *Bos taurus* |
| NC_007704.2 | *Cervus elaphus* | KM093871.1 | *Capra hircus* |
| NC_013840.1 | *Cervus elaphus yarkandensis* | NC_015241.1 | *Microtus fortis fortis* |
| KP405229.1 | *Alces alces cameloides* | KP200876.1 | *Vulpes lagopus* |
| NC_020677.1 | *Alces alces* | HM236180.1 | *Ovis aries* |
| NC_020729.1 | *Odocoileus hemionus* | KT448275.1 | *Canis latrans* |
| NC_015247.1 | *Odocoileus virginianus* | JN632610.1 | *Capreolus capreolus* |
| NC_007703.1 | *Rangifer tarandus* | KJ681493.1 | *Capreolus pygargus* |
| KY987554.1 | *Platygonus compressus* | JN632629.1 | *Dama dama* |
| NC_002008.4 | *Canis lupus familiaris* | KM982549.1 | *Lynx lynx* |
| NC_009686.1 | *Canis lupus lupus* | KP202265.1 | *Panthera pardus* |
| NC_013445.1 | *Cuon alpinus* | NC_026460.1 | *Rhinolophus macrotis* |
| NC_026529.1 | *Vulpes lagopus* | Y07726.1 | *Ceratotherium simum* |
| NC_028302.1 | *Panthera leo* | NC_005089.1 | *Mus musculus* |
| NC_022842.1 | *Panthera onca* | AM711900.1 | *Meles meles* |
| NC_010642.1 | *Panthera tigris* | KM091450.1 | *Mustela erminea* |
| NC_014456.1 | *Lynx rufus* | NC_005358.1 | *Ochotona princeps* |
| NC_020642.1 | *Martes americana* | NC_012095.1 | *Sus scrofa domesticus* |
| NC_020641.1 | *Neovison vison* | DQ480489.1 | *Canis lupus familiaris* |
| NC_024942.1 | *Mustela nigripes* | NC_020670.1 | *Crocuta crocuta* |
| NC_020664.1 | *Martes pennanti* | NC_011116.1 | *Arctodus simus* |
| NC_020639.1 | *Mustela nivalis* | NC_027963.1 | *Sorex araneus* |
| NC_009685.1 | *Gulo gulo* | NC_025327.1 | *Sorex tundrensis* |
| NC_011112.1 | *Ursus spelaeus* | KJ397607.1 | *Lepus arcticus* |
| NC_003426.1 | *Ursus americanus* | NC_001640.1 | *Equus caballus* |
| NC_003427.1 | *Ursus arctos* | NC_024030.1 | *Equus przewalskii* |
| NC_003428.1 | *Ursus maritimus* | | |

**Appendix 1—table 8.** Command lines and software used for initial mapping, processing, and taxonomic assignment.

| | |
|---|---|
| adapter trimming, filtering, and merging: leeHom **Renaud et al., 2014**: | src/leeHom -t 120 –ancientdna –auto -fq1 file_R1.fastq.gz -fq2 file_R2.fastq.gz -fqo leehom_out |
| mapping: BWA **Li and Durbin, 2009** against 75 mammal mitogenomes **Appendix 1—table 3**: | bwa index reference_mitogenomes.fasta<br>bwa aln reference_mitogenomes.fasta leehom_out.fq.gz -l 16,000 n 0.01 -O 2 -o 2 t 8>bwa_out.sai<br>bwa samse reference_mitogenomes.fasta bwa_out.sai leehom_out.fq.gz> bwa_out.sam<br>samtools view -q ≥ 30 S -b bwa_out.sam>bwa_out.bam |
| remove duplicates: samtools **Li and Durbin, 2009** | samtools collate -o bwa_out_col.bam bwa_out.bam<br>samtools fixmate -m bwa_out_col.bam bwa_out_fixmate.bam<br>samtools sort -o bwa_out_pos.bam bwa_out_fixmate.bam<br>samtools markdup bwa_out_pos.bam bwa_out_mark.bam<br>samtools fastq bwa_out_mark.bam>bwa_out.fastq |
| fastq to fasta | sed -n '1~4 s/^@/>/p;2~4 p' bwa_out.fastq>bwa_out.fasta |
| | #alignment: blastn **Altschul et al., 1990**<br>blastn -db ncbi_nt -query bwa_out.fasta -evalue 0.01 -out blastn_out.fasta |
| aDNA damage: mapDamage **Ginolhac et al., 2011** | bwa index reference.fasta<br>bwa aln reference.fasta sample.fasta >sample.sai<br>bwa samse referecne.fasta sample.sai sample.fasta >sample.sam<br>samtools view -q 25 S -b sample.sam >sample.bam<br>mapDamage -i sample.bam -r reference.fasta<br>mapDamage -d results_ sample/ -y 0.1 --plot-only<br>mapDamage -i sample.bam -r reference.fasta –rescale<br>mapDamage -d results_sample / --forward --stats-only -v -r reference.fasta |

## Mammoth mitogenome haplogroup identification

The sequences mapping to mammoth from the libraries with the most mammoth reads were remapped to the mammoth reference (NC_007596) with minimap2 (https://github.com/lh3/minimap2, Copy archieved at **Li, 2024**) to generate a bam file. A panel of 15 mitogenome reference sequences (see **Appendix 1—table 5**) exemplifying the diversity of mammoths was used to identify variants representative of each haplogroup. In cases where there are multiple reference genomes for a haplogroup, the consensus variants unique to the haplogroup were used except for haplogroups B and D&E where haplosubgroups were also identified. mixemt (**Vohr et al., 2017**, https://github.com/svohr/mixemt) was then used to detect haplogroups present in the core and estimate the mixture proportions of these haplogroups, using only mapped reads with a map quality of >30 and with the requirement that 40% of the unique defining variants of each haplogroup must be observed to say it is present.

**Appendix 1—table 9.** Reference mammoth mitogenomes used for panel.

| Clade | Haplogroup | Subgroup | Accession numbers | Proportion of reads assigned to haplogroup |
|---|---|---|---|---|
| II | A | | EU153451, EU153450 | - |
| III | B | B0 | KX027526, KX027531 | - |
| | | B1 | KX027526 | 0.1615 |
| | | B2 | KX027531 | - |
| I | C | | KX027498, KX027565, KX027567, JF912200, KX027499, KX027502 | - |

*Appendix 1—table 9 Continued on next page*

*Appendix 1—table 9 Continued*

| Clade | Haplogroup | Subgroup | Accession numbers | Proportion of reads assigned to haplogroup |
|---|---|---|---|---|
| | | D0 | DQ316067, EU153454, EU153447, EU153456, EU153449, EU153455, EU153446 | - |
| | | D1 | DQ316067 | - |
| | | D2 | EU153454 | - |
| I | D&E | D3 | EU153447 | 0.2790 |
| | | D4 | EU153456 | - |
| | | D5 | EU153449 | 0.0765 |
| | | D6 | EU153455 | 0.4829 |
| | | D7 | EU153446 | - |
| I | F | | KX027503, KX027512, NC_015529, KX027511, KX027548, KX027547, KX027556, KX027559, KX027550 | - |
| Krestovka mammoth | K | | PRJEB42269 (European Nucleotide Archive) | - |

| Conventional PCR | | | |
|---|---|---|---|
| 94 °C | 2 min | | |
| 94 °C | 30 s | | |
| 54 °C | 30 s | 55 cycles | |
| 68 °C | 20 s | | |
| 68 °C | 1 min | | |

## Conventional PCR

The PCR reaction mix comprised 17.05 μL $H_2O$, 0.2 μL Platinum Taq DNA-Polymerase High Fidelity (Thermo Fisher Scientific, Waltham, MA, USA), 2.5 μL 10 X reaction buffer, 0.25 μL dNTPs (25 mM), 1 μL bovine serum albumin (20 mg/mL; New England Biolabs, Ipswitch, MA, USA), 1 μL $MgSO_4$ (50 mM), 1 μL of each primer (mamm801 and mamm800r; 10 μM), and 1 μL DNA extract. Thermocycling was performed as shown in *Appendix 1—table 9*.

Amplification products were visualized by gel electrophoresis and were purified from excised gel bands using a NucleoSpin Gel and PCR Clean-up kit (Macherey-Nagel, Düren, Germany), followed by Sanger sequencing.

## Mammal metabarcoding

We used the primers MamP007F and MamP007R (*Giguet-Covex et al., 2014*) and blockers (*Garcés-Pastor et al., 2021*), with the following conditions:

**Appendix 1—table 10.** Conventional PCR.

| Reagent | μL |
|---|---|
| $H_2O$ | 14.85 |
| Platinum Hifi Taq Buffer 10 X (Thermo Fisher Scientific) | 2.5 |
| bovine serum albumin (New England Biolabs) | 0.2 |
| $MgSO_4$ (50 mM) | 1.00 |
| dNTPs (25 mM) | 0.25 |
| Platinum HiFi (5 U/μL; Thermo Fisher Scientific) | 0.20 |

*Appendix 1—table 10 Continued on next page*

*Appendix 1—table 10 Continued*

| Reagent | µL | |
|---|---|---|
| Blocking primer R (1 µM) | 0.5 | |
| Blocking primer F (1 µM) | 0.5 | |
| total | 20.00 | |
| template DNA | 3.00 | |
| **Thermocycling** | | |
| °C | t | |
| 94 | 5 min | |
| 94 | 30 s | 40 cycles |
| 50 | 30 s | |
| 68 | 30 s | |
| 68 | 10 min | |
| RT | store | |

PCR products were purified using a MinElute kit (Qiagen). Sequencing was performed on an Illumina NovaSeq platform, with 2x150 reads. The raw data were processed as described below.

## Bioinformatics of mammal metabarcoding data

The obtained datasets by Illumina Sequencing were then processed as follows:

**Appendix 1—table 11.** Conventional PCR.

| | |
|---|---|
| *Load data* | obi import --quality-sanger file_R1.fastq reads1<br>obi import --quality-sanger file_R2.fastq reads2 |
| *Import tags* | obi import --ngsfilter-input taglist.txt ngsfile |
| *Align paired-end reads* | obi alignpairedend -R reads2 reads1 aligned_reads |
| *Grep entries whose mode are alignment* | obi grep -a mode:alignment aligned_reads good_sequences |
| *Assign alignments to individual PCRs* | obi ngsfilter -t ngsfile -u unidentified_sequences good_sequences identified_sequences |
| *Filter out sequences* | obi grep -p "sequence['score']>50" identified_sequences identified_sequences_filtered<br>obi grep -p "sequence['score_norm']>0.9" identified_sequences_filtered identified_sequences_filtered_adj |
| *Dereplicate Sequences* | obi uniq -m sample identified_sequences_filtered_adj dereplicated_sequences_filtered |
| *Keep only useful tags* | obi annotate -k COUNT -k MERGED_sample dereplicated_sequences_filtered cleaned_metadata_sequences |
| *Discard sequences that are shorter than 60 bp (based on primer pair)* | obi grep -p "len(sequence) ≥ 60 and sequence['COUNT'] ≥ 10" cleaned_metadata_sequences denoised_sequences |
| *Clean the sequences from PCR/sequencing errors* | obi clean -s MERGED_sample -r 0.05 H denoised_sequences cleaned_sequences |
| *Load database* | cp STD_MAM_1.dat.gz ~/edna_LauraB/master/mammalia/database/ |
| *Import it into DMS* | obi import /data/scc/edna/LauraBa/master/mammalia/database/STD_MAM_1.dat.gz database_mam<br>obi import --embl EMBL embl_refs |
| *Download the taxonomy* | wget https://ftp.ncbi.nlm.nih.gov/pub/taxonomy/taxdump.tar.gz |
| *Import the taxonomy in the DMS* | obi import --taxdump /data/scc/edna/LauraBa/master/mammalia/taxdump.tar.gz taxonomy/my_tax |

*Appendix 1—table 11 Continued on next page*

*Appendix 1—table 11 Continued*

| | |
|---|---|
| *Cleaning the database with in silico PCR* | obi ecopcr -e 3 l 50 L 150 F CGAGAAGACCCTATGGAGCT -R CCGA GGTCRCCCCAACC --taxonomy taxonomy/my_tax embl_refs mam_ refs |
| *Filter sequences* | obi grep --require-rank=species --require-rank=genus --require-rank=family --taxonomy taxonomy/my_tax mam_refs mam_refs_clean |
| *Dereplicate identical sequences* | obi uniq --taxonomy taxonomy/my_tax mam_refs_clean mam_refs_uniq |
| *Add taxid at the family level* | obi grep --require-rank=family --taxonomy taxonomy/my_tax mam_ refs_uniq |
| *Build the reference database* | obi build_ref_db -t 0.97 --taxonomy taxonomy/my_tax mam_refs_ uniq_clean mam_db_97 |
| *Assign each sequence to a taxon* | obi ecotag -m 0.97 --taxonomy taxonomy/my_tax -R mam_db_97 cleaned_sequences assigned_sequences |
| *Align the sequences* | obi align -t 0.95 assigned_sequences aligned_assigned_sequences |
| *Export tables for downstream data analysis* | obi grep -A SCIENTIFIC_NAME assigned_sequences assigned_for_metabR |
| *Output two tables required by metabaR* | obi annotate -k MERGED_sample assigned_for_metabR assigned_ for_metabR_reads_tableobi export --tab-output --output-na-string 0 assigned_for_metabR_reads_table >mam_reads_01.txtobi annotate --taxonomy taxonomy/my_tax \<br>--with-taxon-at-rank superkingdom \<br>--with-taxon-at-rank kingdom \<br>--with-taxon-at-rank phylum \<br>--with-taxon-at-rank subphylum \<br>--with-taxon-at-rank class \<br>--with-taxon-at-rank subclass \<br>--with-taxon-at-rank order \<br>--with-taxon-at-rank suborder \<br>--with-taxon-at-rank infraorder \<br>--with-taxon-at-rank superfamily \<br>--with-taxon-at-rank family \<br>--with-taxon-at-rank genus \<br>--with-taxon-at-rank species \<br>--with-taxon-at-rank subspecies \ assigned_for_metabR assigned_for_ metabR_taxInfo<br>obi annotate \<br>-k BEST_IDENTITY -k TAXID -k SCIENTIFIC_NAME -k COUNT -k seq_length \<br>-k superkingdom_name \<br>-k kingdom_name \<br>-k phylum_name \<br>-k subphylum_name \<br>-k class_name \<br>-k subclass_name \<br>-k order_name \<br>-k suborder_name \<br>-k infraorder_name \<br>-k superfamily_name \<br>-k family_name \<br>-k genus_name \<br>-k species_name \<br>assigned_for_metabR_taxInfo assigned_for_metabR_motus<br>obi export --tab-output assigned_for_metabR_motus >mam_motus_ 01.txt |
| Further processing of the data sets was done using RStudio. | |
| *Editing files for metabar* | reads<- dt_reads %>%<br>dplyr::select(-c("DEFINITION", "NUC_SEQ"))%>% as.data.frame() %>%<br>janitor::row_to_names(row_number = 897, remove_rows_above = FALSE, remove_row = TRUE) %>% mutate_if(is.integer,as.numeric) |

*Appendix 1—table 11 Continued on next page*

*Appendix 1—table 11 Continued*

| | |
|---|---|
| *Assign name to first column* | reads <- cbind(rownames(reads),reads)<br>rownames(reads) <- NULL<br>colnames(reads) <- c(names(reads))<br>colnames(reads)(1) <- "pcr_id" |
| *Edit the names of the column* | reads$pcr_id = strsplit(reads$pcr_id,"[.]") reads$pcr_id =<br>sapply(reads$pcr_id, function(x) x[length(x)]) rownames(reads)<-<br>reads$pcr_id |
| *Organizing the MOTUs table* | motus<- dplyr::select(dt_motus, 'ID', 'NUC_SEQ', 'COUNT','BEST_<br>IDENTITY', 'TAXID', 'SCIENTIFIC_NAME', 'superkingdom_name',<br>'species_name', 'class_name', 'order_name', 'family_name',<br>'genus_name', 'kingdom_name', 'phylum_name', 'subphylum_name',<br>'subclass_name', 'suborder_name')<br>names(motus)[names(motus) == 'NUC_SEQ'] <- 'sequence' |

## Droplet digital PCR (ddPCR)

ddPCR was performed using a Bio-Rad QX200 system (Bio-Rad Hercules, CA, USA) using the following conditions, and data were produced and analyzed as per the standard instructions of the manufacturer. Droplets were generated by using 21 µL of the PCR mixture and 70 µL ddPCR Droplet Reader Oil (Bio-Rad) according to the manufacturer's instructions. The final volume for PCR amplification was 40 µL and was carried out in a C1000 Touch Thermo cycler (Bio-Rad). The products were analyzed using a QX200 Droplet Reader (Bio-Rad); the threshold was set manually to 3000.

**Appendix 1—table 12.** Conventional PCR.

| Reagent | µL | |
|---|---|---|
| ddPCR Supermix for probes | 11 | |
| H2O (DEPC) | 6.8 | |
| 20 x Target-Primers/Probe (FAM) | 1.1 | |
| 20 x Target-Primers/Probe (HEX) | 1.1 | |
| total | 20 | |
| template DNA | 2.0 | |
| **Thermocycling** | | |
| °C | t | |
| 95 | 10 min | |
| 94 | 30 sec | |
| 50 | 30 sec | 40 cycles |
| 60 | 30 sec | |

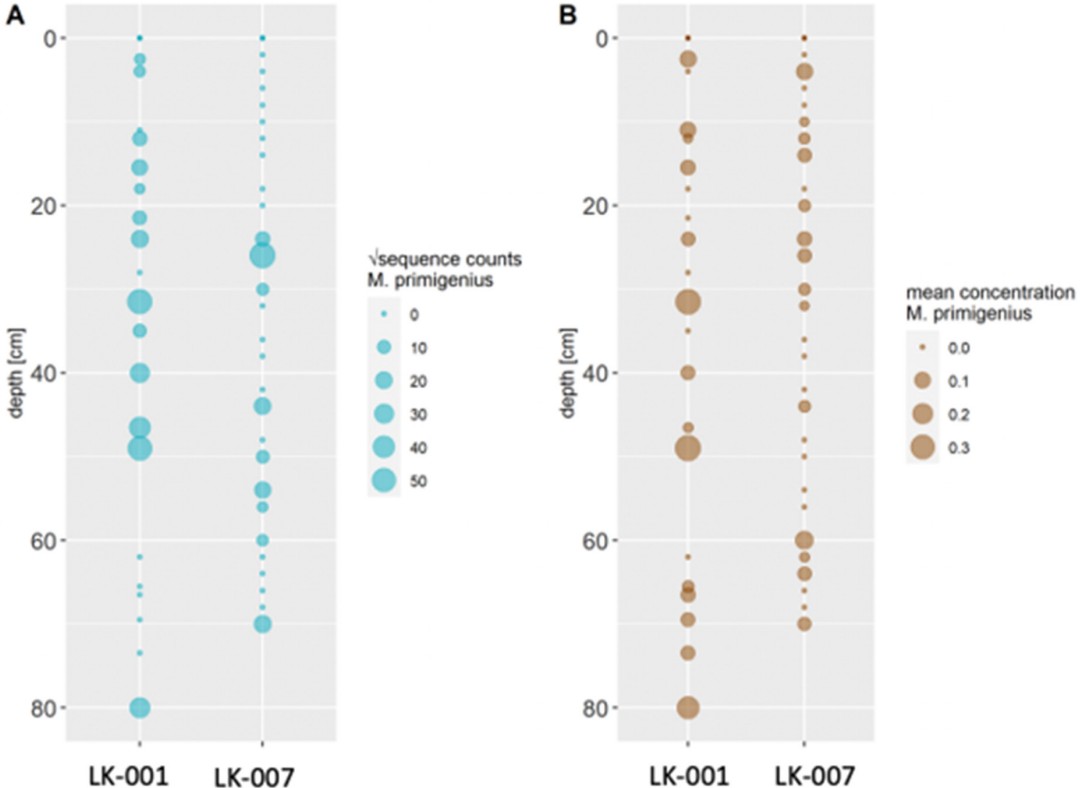

**Appendix 1—figure 1.** Comparison of transformed metabarcoding read counts (**A**) and ddPCR concentration estimations (**B**) of *Mammuthus primigenius* DNA in sediment cores LK-001 and LK-007 from the Yamal peninsula.

## Sediment dating

The dating sections of core LK-001 (***Appendix 1—table 13***) were dried to constant weight at 65 °C and were then analyzed for $^{210}$Pb, $^{226}$Ra, and $^{137}$Cs by direct gamma assay at the Liverpool University Environmental Radioactivity Laboratory using Ortec HPGe GWL series well-type coaxial low background intrinsic germanium detectors. This allowed dating to a depth of 39.5 cm (***Appendix 1—table 13***).

**Appendix 1—table 13.** $^{210}$Pb chronology of Yamal lake sediment core LK-001.

| Depth | | Chronology | | | Sedimentation Rate | |
|---|---|---|---|---|---|---|
| | | Date | Age | | | |
| cm | g cm$^{-2}$ | AD | y | ± | g cm$^{-2}$ y$^{-1}$ | cm y$^{-1}$ |
| 0.00 | 0.0 | 2019 | 0 | 0 | | |
| 0.25 | 0.2 | 2018 | 1 | 1 | 0.34 | 0.37 |
| 1.25 | 1.1 | 2016 | 3 | 2 | 0.34 | 0.37 |
| 2.25 | 1.9 | 2013 | 6 | 2 | 0.34 | 0.37 |
| 3.25 | 2.8 | 2010 | 9 | 2 | 0.34 | 0.37 |
| 4.25 | 3.8 | 2008 | 11 | 2 | 0.34 | 0.37 |
| 5.25 | 4.7 | 2005 | 14 | 2 | 0.34 | 0.37 |
| 6.25 | 5.7 | 2002 | 17 | 2 | 0.34 | 0.37 |
| 7.25 | 6.7 | 1999 | 20 | 3 | 0.36 | 0.37 |
| 8.25 | 7.6 | 1996 | 23 | 3 | 0.39 | 0.41 |

*Appendix 1—table 13 Continued on next page*

*Appendix 1—table 13 Continued*

| Depth | | Chronology | | | Sedimentation Rate | |
|---|---|---|---|---|---|---|
| 10.25 | 9.7 | 1992 | 27 | 3 | 0.49 | 0.47 |
| 12.25 | 11.8 | 1988 | 31 | 4 | 0.53 | 0.49 |
| 14.25 | 14.0 | 1984 | 35 | 4 | 0.55 | 0.51 |
| 14.75 | 14.5 | 1983 | 36 | 4 | 0.55 | 0.51 |
| 15.50 | 15.4 | 1981 | 38 | 4 | 0.55 | 0.51 |
| 16.50 | 16.5 | 1980 | 39 | 4 | 0.55 | 0.51 |
| 17.50 | 17.6 | 1978 | 41 | 4 | 0.55 | 0.51 |
| 18.50 | 18.6 | 1976 | 43 | 4 | 0.55 | 0.51 |
| 20.50 | 20.7 | 1971 | 48 | 5 | 0.55 | 0.51 |
| 22.50 | 22.7 | 1968 | 51 | 6 | 0.55 | 0.51 |
| 23.50 | 23.7 | 1966 | 53 | 6 | 0.55 | 0.51 |
| 24.50 | 24.8 | 1964 | 55 | 6 | 0.55 | 0.51 |
| 25.50 | 25.9 | 1962 | 57 | 7 | 0.44 | 0.42 |
| 26.50 | 27.0 | 1961 | 58 | 7 | 0.37 | 0.36 |
| 27.50 | 28.0 | 1957 | 62 | 8 | 0.30 | 0.29 |
| 28.50 | 29.0 | 1952 | 67 | 9 | 0.28 | 0.22 |
| 29.50 | 30.2 | 1949 | 70 | 10 | 0.23 | 0.19 |
| 31.50 | 32.6 | 1938 | 81 | 10 | 0.23 | 0.19 |
| 33.50 | 34.7 | 1933 | 86 | 10 | 0.23 | 0.19 |
| 35.50 | 36.9 | 1927 | 92 | 11 | 0.23 | 0.19 |
| 37.50 | 39.1 | 1913 | 106 | 13 | 0.23 | 0.19 |
| 39.50 | 41.4 | 1895 | 124 | 17 | 0.23 | 0.19 |

Core LK-001 was additionally dated using radiocarbon. We collected plant remains throughout the core by first using a scalpel to cut 1 cm thick of bulk sediment at desired depths, followed by washing the sediment chunks with distilled water and filtering using 190 µm mesh sieves. We then collected leaves of deciduous trees and seeds, which were placed in glass vials and dried under a fume hood before sending to the Micadas (Alfred Wegener Institute, Bremerhaven, Germany) for radiocarbon dating. Sample preparation and measurement methodologies at the Micadas were performed as described previously (*Mollenhauer et al., 2021*). We sent three samples that had sufficient plant materials for dating and retained two dates after removing an inaccurate date caused by high inbuilt age from plant remains (*Appendix 1—table 14*).

**Appendix 1—table 14.** Radiocarbon dating results.

| Sample label | F14C | ± (abs) | Age (y) | ± (y) | Weight (µg C) | Comment |
|---|---|---|---|---|---|---|
| LK-001_36.5 cm | 0.9595 | 0.0095 | 332 | 79 | 35 | |
| LK-001_51 cm | 0.3396 | 0.0056 | 8677 | 132 | 148 | Off; lateral input |
| LK-001_74 cm | 0.8248 | 0.0234 | 1547 | 228 | 13 | |

# Mammal read numbers
The numbers of reads assigned to all mammals are indicated in *Appendix 1—table 15* (core LK-001 only; hybridization capture experiment).

**Appendix 1—table 15.** Numbers of sequences assigned to mammals (core LK-001).

| | sum | depth 1.5 | 4.0 | 7.0 | 11.0 | 12.0 | 15.5 | 18.0 | 21.5 | 23.0 | 28.0 | 31.5 | 35.0 | 40.0 | 46.5 | 49.0 | 51.0 | 62.0 | 65.5 | 66.5 | 69.5 | 73.5 | 80.0 |
|---|---|---|---|---|---|---|---|---|---|---|---|---|---|---|---|---|---|---|---|---|---|---|---|
| *Mammuthus primigenius* | 19,640 | 191 | 162 | 524 | 290 | 1205 | 605 | 194 | 666 | 277 | 800 | 3697 | 305 | 1188 | 1374 | 1923 | 631 | 674 | 519 | 1110 | 1083 | 814 | 1408 |
| *Rangifer tarandus* | 18,055 | 253 | 63 | - | 34 | 328 | 688 | 644 | 1816 | 545 | 902 | 2186 | 1240 | 1571 | 1574 | 86 | 284 | 2714 | 56 | 345 | 991 | 114 | 1,621 |
| *Dicrostonyx torquatus* | 16,870 | 211 | 112 | 34 | 111 | 1211 | 614 | 271 | 1035 | 522 | 1049 | 850 | 1298 | 314 | 761 | 193 | 318 | 656 | 200 | 2763 | 1011 | 1408 | 1928 |
| *Lepus* | 6371 | 148 | 134 | 105 | 23 | 584 | 245 | 40 | 369 | 336 | 630 | 382 | 231 | 404 | 799 | 285 | 141 | 116 | 126 | 352 | 330 | 293 | 298 |
| *Coelodonta antiquitatis* | 2737 | 33 | - | - | 28 | 119 | 153 | 347 | 55 | - | 97 | 128 | 467 | 298 | 141 | 27 | 103 | 152 | - | 250 | 144 | 4 | 191 |
| *Homo sapiens* | 387 | - | - | - | - | - | - | 38 | 217 | - | - | 23 | - | 25 | - | 69 | - | 15 | - | - | - | - | - |
| *Ovibos moschatus moschatus* | 145 | - | 69 | - | - | - | - | - | - | - | 38 | - | - | - | 38 | - | - | - | - | - | - | - | - |
| *Castor fiber* | 135 | - | - | - | - | 18 | - | - | - | - | 15 | - | - | - | 27 | 51 | - | 6 | 1 | - | - | - | 17 |
| *Bos* | 105 | - | - | - | - | - | - | - | - | - | - | - | - | - | - | 105 | - | - | - | - | - | - | - |
| *Cervinae* | 92 | - | - | - | - | - | - | - | 33 | - | - | - | - | 31 | 28 | - | - | - | - | - | - | - | - |
| *Saiga tatarica* | 91 | - | - | - | - | - | - | - | - | - | 57 | - | 34 | - | - | - | - | - | - | - | - | - | - |
| *Sus scrofa cristatus* | 85 | - | - | - | - | - | - | - | 34 | - | - | - | - | - | - | 24 | - | - | - | - | 27 | - | - |
| *Ochotona* | 74 | - | - | - | - | - | - | - | - | - | - | - | - | - | - | - | - | - | 14 | - | - | - | 60 |
| *Ovis aries musimon* | 70 | - | - | - | - | 70 | - | - | - | - | - | - | - | - | - | - | - | - | - | - | - | - | - |
| *Bison* | 54 | - | - | - | - | 24 | - | 30 | - | - | - | - | - | - | - | - | - | - | - | - | - | - | - |
| *Crocuta crocuta* | 53 | - | - | - | - | - | - | - | - | - | - | - | - | - | - | - | - | - | - | 53 | - | - | - |
| *Chrotopterus auritus* | 43 | - | - | - | - | - | 43 | - | - | - | - | - | - | - | - | - | - | - | - | - | - | - | - |
| *Sorex tundrensis* | 32 | - | - | - | - | - | - | - | - | - | - | - | - | - | - | - | 32 | - | - | - | - | - | - |
| *Equus* | 32 | - | - | - | - | - | - | - | - | - | - | - | - | - | - | - | - | - | 7 | - | - | - | 25 |
| *Peromyscus maniculatus bairdii* | 29 | - | - | - | - | - | - | - | - | - | - | - | - | - | 29 | - | - | - | - | - | - | - | - |
| *Murinae* | 28 | - | - | - | - | - | - | - | - | - | - | - | - | - | - | - | - | - | - | - | - | 28 | - |
| *Capra* | 21 | - | - | - | - | - | - | - | - | - | - | - | - | - | - | - | 21 | - | - | - | - | - | - |
| *Myopus schisticolor* | 18 | - | - | - | - | - | - | - | 18 | - | - | - | - | - | - | - | - | - | - | - | - | - | - |
| *Lemmus trimucronatus* | 16 | 16 | - | - | - | - | - | - | - | - | - | - | - | - | - | - | - | - | - | - | - | - | - |

