## [Editor Report · eLife assessment]

This work presents **convincing** evidence for the presence of wooly mammoth/rhinoceros ancient environmental DNA (aeDNA) far from the time likely to host living individuals: what is effectively a genetic version of a geological inclusion. These are **important** findings that will have ramifications for the interpretation and conclusions extracted from aeDNA more generally.

---

## [Referee Report · Reviewer #2 (Public Review)]

Summary:

The authors report the successful retrieval of mitogenomes from extinct Pleistocene megafauna (woolly Mammoth and woolly rhino) from recent sediment cores from two close Siberian lakes. The cores are too recent to represent real time points of these two extinct species (known to have been extinct for several thousands of years) and therefore, the most plausible interpretation is that permafrost thawing and similar physical processes in the lakes have made surface old ancient DNA, maybe from nearby, deep-buried carcasses.

They have answered the comments and questions I raised in my review. I agree with them on the complexities or separating a potential mixing of different Mammoth mito genomes retrieved.

---

## [Referee Report · Reviewer #3 (Public Review)]

Summary:

In this study, the researchers used ancient environmental DNA (aeDNA) retrieved from sediment cores, from two lakes in the Arctic, on the Yamal peninsula, in Siberia. The dating of one of the cores, showed that the sediment layers were very recent (ranging between the years 2019 - 1895). From this core they sequenced 23 libraries which were enriched for mammal mitochondrial genomes. They found a high proportion of two species that have been extinct for thousands of years, the mammoth and the woolly rhinoceros. The highest proportion of mammoth reads were found in very young layer (~81 years old) and as this initial finding does not match the temporal occurrence of the species, they confirmed the identification with several other methods. Additionally, they applied a different dating method on some samples and found that the aging of the samples was not completely congruent. The authors suggest the that the presence of these two Pleistocene megafauna in such recent sediment layers is a consequence of physical processes, specific to the study site, and that the high quality of the aeDNA recovered is a result of permafrost preservation.

The strengths of the study are in the rigorous confirmation of the identification of the taxa with four different PCR and sequencing techniques being used, the initial enrichment panel, and then subsequent metabarcoding PCRs, and taxa specific PCR for COI and cytB. Along with the ancient DNA protocol applied, this is therefore very convincing that the DNA detected in the samples is indeed from the Pleistocene mammals. Additionally, two methods were used to age the sediment cores, and although the depth of the samples tested do not overlap, they give reasonable ages (apart from the anomalous sample) and all together these are robust results.

There is now an analysis supporting the idea that there are multiple individual mammoths in the sample as well as a figure to display the locations of the haplotypes. The authors also confirm that the woolly rhinocerous did not recover enough sequences for analysis. The aims have been clarified and no longer states that they are looking at mammal biodiversity through time, so the papers focus is now more specifically on just the mammoth. But a supplementary table of the reads from common mammals has been added.

Overall the results support that there has been some movement of DNA throughout the sediment core which may impact the dating of the last occurrence of particular extinct taxa. As highlighted, though the geological processes by which this may have arisen are specific to this particular lake and may not be broadly relevant, therefore highlighting that knowledge of each system is important to understanding DNA distribution.

---

## [Author Response]

The following is the authors’ response to the original reviews.

The reviewers make some suggestions aimed towards increasing the clarity of the manuscript, and I suggest that the authors examine those carefully. In particular, the figure is difficult to read and could contain additional information to help the reader's interpretation. For example, Reviewer 1 suggests including sample age estimates alongside depth, while Reviewer 3 also notes that there is missing information in the figure. Apart from the figure, Reviewer 1 suggests two additional analysis to help explain the amount of mammoth DNA recovered, which they observe is much higher than previous similar investigations. This would seem to be an important issue to address, given the surprising nature of the findings. In addition to this larger issue, the Reviewer makes a few important suggestions for supplementary material that may be needed to support the authors' statements.Some additional recommended edits -- in particular to the text and included references to related studies -- are suggested by Reviewers 2 and 3, and both commented on the lack of a publicly-available data repository. The authors may also wish to comment on or revisit their differential treatment of wooly mammoth vs. wooly rhinoceros samples, though I suspect this has more to do with low read numbers for the rhinos.

Thank you very much for the positive assessment of our manuscript and clear suggestions for revision. We address these points below.

**Reviewer #1 (Recommendations For The Authors):**
I have a few suggestions that might further improve the manuscript:It is difficult for the reader to follow which core slices exactly have been sampled and sequenced. The authors mention 23 samples were taken from core LK-001 and 16 samples from core LK-007. From the text it remains unclear to me what the exact age of each of these samples is. Figure 1 shows the depth at which the LK-001 core was sampled, maybe sample age estimates could be included here.

Thanks for pointing this out. We have added approximate ages to Figure 1, added the depth range to the text (“from 1.5 to 80 cm”; l. 73-74, caption Figure 1), and reworked the table of the sampling depths in the supplement.

Line 84-87. The authors mention the retrieval of DNA from several expected Arctic taxa, however no further data regarding these findings is given in the manuscript. It would be useful to report the same numbers for these species as the ones given for the Mammuthus and woolly rhinoceros, which would allow for a comparison of the relative abundance of the DNA between these species. Are the expected Arctic species for instance at much higher (DNA) abundance in the samples? It would also be interesting to know if the authors discovered DNA from extant species that are unlikely to have occurred in the geographic region. A (supplementary)table listing the number of mapped reads to each of the respective mitogenomes for each sequence library would be useful for the reader.

We added a supplementary table (S8) indicating the numbers of reads assigned to mammals.

Line 90: I am somewhat amazed by the amount of mammoth DNA the authors recovered from these cores. A total depth of over 400X of the mitogenome is quite extraordinary and I am not aware of any ancient sediment study to date that has retrieved a similar amount of data. For instance, the Wang et al. 2021 paper, which the authors cite, sequenced over 400 samples and did not find any mammoth DNA in 70% of those. For the 30% of samples showing signs of mammoth DNA they retrieved on average 530 sequence reads. In this study the authors find on average ~20.000 reads, in 22 out of the 23 sequence libraries. This makes me wonder if the way the mapping was performed has been too lenient, resulting in possible spurious mappings? To really confirm the authenticity of the mammoth (and woolly rhino data) I would suggest two additional analysis:1. Mapping all the sequence libraries to a reference consisting of the complete Asian-elephant genome (for instance https://www.ncbi.nlm.nih.gov/datasets/genome/GCF_024166365.1/), the complete human genome (+mitogenome) and the Asian elephant mitogenome. This could possibly reduce spurious mappings as conserved regions between the genomes are filtered out and could also reduce the possible mapping of NUMTS. If the authors could show that after such a mapping approach a significant number of reads are still assigned to the Asian elephant part (including the mitogenome) of the reference, the reported findings would be strengthened.1. I also suggest to construct a mitochondrial haplotype network from the obtained DNA, while also including previously published Asian and African elephants as well as previously published mammoth mitogenomes. If the obtained haplotypes indeed show that they cluster within the known haplotype diversity of mammoth, that would be strong support for the authenticity of the dataThe same analysis could be considered for the woolly rhino data, although the lower read numbers might make this analysis challenging.

We agree that the amount of mammoth DNA is surprising, which is why we opted for further laboratory experiments for confirmation of the hybridization capture results of the first core, i.e., (1) DNA extraction from a second core of a different lake, (2) a quantitative PCR approach (ddPCR), and (3) metabarcoding. Our results of the highly specific ddPCR and metabarcoding assays confirmed considerable amounts of mammoth DNA in two sediment cores of different lakes, thus we have no doubts regarding the authenticity of the data. Considering the large amount of mammoth DNA, the high number of reads, and particularly the high mitogenome coverage, we argue that the effect of some spurious mapping is negligible and does not affect the main outcome and conclusions of our study. Although we agree that a haplotype network would be interesting, such analyses would stretch beyond the focus of this publication.

Line 91: The authors mention negative controls (extraction and library blanks) did not produce any reads assigned to mammals. This is quite remarkable, as in my experience low levels of (human)contamination are almost always present in the blanks. Could the authors comment on why they think the blanks did not show any signal of mammalian DNA?

The hybridization capture enrichment and the filtration and mapping procedures likely eliminated human contamination. Also, the data were mapped against Arctic mammal mitogenomes, which did not include human reference sequences. However, six of the sediment samples contained human sequences (now shown in supplementary table S8), albeit at low read counts (mean = 65)

Line 97: "mapping suggested that the sequences throughout the core originated from multiple individuals" The authors do not provide any supporting data showing this. I think that an analysis (for instance based on allele frequencies) has to be included in manuscript to support this claim.

We agree that his claim was not sufficiently supported. We performed further analyses including genomic data of previously retrieved mammoth remains and assigned our data to these haplogroups; the results were added to the main text and are shown as a figure (Fig. 2).

Line 98: "Signatures of post-mortem DNA decay were comparably minor."Do the authors know if the used hybridisation enrichment method can distort the measurement of post-mortem damage? Are for instance reads with C-T substitutions less likely to be captured by the baits?

To our knowledge, there is no study suggesting that damaged sites are less likely to be captured. In general, the hybridization capture procedure is not overly specific, and studies report that DNA is readily and preferentially captured as long as the difference between baits and DNA is not above 10%.

Line 100: "The proportions of bases did not suggest a substantial deviation from those in the reference genomes or in the closest extant relative of Mammuthus, the Asian elephant (Elephas maximus)."It is not clear to me what the authors mean by this. Could the authors explain how this was measured and what their interpretation of this result is?

We realize that the sentence was unclear. We meant that the nucleotide composition was similar to that of the reference genomes or the closest extant relative. However, as we do not consider this important for the argument, we have removed this sentence from the manuscript.

Given the high number of recovered mammoth reads in the samples, it would be interesting to know how much mammoth reads are present in the sample before enrichment capture with the baits. Shotgun sequencing the raw extract of one of the samples with the highest number of mammoth reads might allow for a rough estimate of mammoth DNA abundance compared to the other extant species (e.g. reindeer, Arctic lemming and hare) found in the sample(s). This could give further clarification about the extent of stratigraphy disturbance and its overall effect on the DNA based community reconstruction. However, this is just a suggested additional analysis and not something I believe crucial for supporting the overall findings in this manuscript.

We fully agree that this would be a highly interesting and informative additional analysis to perform. It was, however, not possible to perform this additional analyses in the course of the current experiments.

Finally, I could not find a public link to the (sequence)data produced in this study. I strongly encourage the authors to make their data publicly available.

Thank you for pointing this out. We have added a Data Availability paragraph, including the respective reference.

**Reviewer #2 (Recommendations For The Authors):**
In the Discussion it is mentioned that the reasons for Mammoth extinction are not entirely clear but are largely attributed to sudden climate warming (and add some relevant citations). However, there is also abundant literature that suggest humans also played a role in their extinction (for instance, a recent one, Damien et al. (2022) at Ecology Letters 25: 127-137).

We agree with the reviewer and have added some the recent citation highlighting the possible influence of humans.

One possibility to add further interest to this paper would be to conduct a phylogenetic tree with the Mammoth mitogenome(s) retrieved and a reference dataset; it could be interesting to know where do they fall in the phylogeny -already abundant with tens of individuals- and maybe it could be even possible to roughly estimate their date. There are some papers that report many Mammoth mitogenomes, including of course some from Siberia; for instance Chang et al. (2017) at Sci Reports and also Fellow Yates et al. (2017) also at Sci Reports (the latter mainly from Central Europe).

We are well aware of the amount of mt genomes available for mammoth, and such an analyses would be an interesting addition, potentially also offering the possibility to date the DNA. However, the analyses was hampered and would be less secure for this dataset, as our sequences display quite some variation among each other, suggesting that we have a mix of multiple mt genomes, which we cannot readily distinguish. We thus refrain from this, also because we instead provide multiple lines of evidence for the existence of the mammoth DNA in the surface sediment core (metabarcoding, ddPCR).

Minor points:-Correct wooly to woolly

Revised.

-In the sampling description it is not totally clear if the samples were taken at 1 cm each (it is mentioned that core LK-001 is sliced in the field at 1-cm steps for radiometric dating and later it is explained that 23 samples were analyzed from this core, but it is unclear if they represent 23 cm of core)-Maybe the authors could briefly define some terms such as "talik"

Revised.

**Reviewer #3 (Recommendations For The Authors):**
Maybe I missed this but I could not find a data availability statement or the location of the repository

We have added a Data Availability paragraph, including the respective reference.

It would be good to see some additional analysis on the distribution of the woolly rhinoceros DNA through the sediment core - like the figure for the mammoth i.e read numbers vs depth.

We have added to the supplements a table showing the numbers of assigned mammal reads over the core depths (Table S8). However, as rhinoceros reads are considerable rarer in our results, we did not produce a figure.

Would it be possible to be more explicit about the multiple mammoth individuals, could you calculate a minimum number or haplotypes for example.

We agree that his claim was not sufficiently supported and added results from additional analyses (incl. Fig. 2). Please see our response above.

Based on the aim stated in the introduction, the analysis of the Arctic biodiversity of this area is missing, it would be nice to see these result added or maybe the focus needs to be changed for clarity.

We now explicitly state that this objective pertains to a different study, which is currently still in preparation for publication.

The single main figure needs a bit more consideration. For example in panel A - there was no information on the transformation performed or what the general trend line refers to. Do the results in panel B refer to all 22 libraries? What is the x-axis in Panel C and what do the coloured lines refer to? Additionally, I think the figure needs to be in higher resolution with increased text size on all axes.

We revised the figure and the caption for clarity and readability.

Finally this might be an accidental typo - but when referring to the sample aged at around 8,677 years in text it states this the 36.5 cm sample (line 130 and 192), but the supplementary says this is the 51cm sample (Table S6). This would maybe impact potential conclusions. Would you be able to clarify this.

Thank you for noting this error, we revised it.